


**The impact of drought on the productivity of two rainfed crops in Spain**
Marina Peña-Gallardo[1], Sergio Martín Vicente-Serrano[1], Fernando Domínguez-Castro[1], Santiago Beguería[2]
[1] Instituto Pirenaico de Ecología, Consejo Superior de Investigaciones Científicas (IPE-CSIC), Zaragoza, Spain.
[2] Estación Experimental de Aula Dei, Consejo Superior de Investigaciones Científicas (EEAD-CSIC), Zaragoza, Spain.
**Abstract**
Drought events are of great importance in most Mediterranean climate regions because of the diverse and costly impacts they
have in various economic sectors and on the environment. The effects of this natural hazard on rainfed crops are particularly
evident. In this study the impacts of drought on two representative rainfed crops in Spain (wheat and barley) were assessed.
As the agriculture sector is vulnerable to climate, it is especially important to identify the most appropriate tools for monitoring
the impact of the weather on crops, and particularly the impact of drought. Drought indices are the most effective tool for that
purpose. Various drought indices have been used to assess the influence of drought on crop yields in Spain, including the
standardized precipitation and evapotranspiration index (SPEI), the standardized precipitation index (SPI), the Palmer drought
indices (PDSI, Z-Index, PHDI, PMDI), and the standardized Palmer drought index (SPDI). Two sets of crop yield data at
different spatial scales and temporal periods were used in the analysis. The results showed that drought indices calculated at
different time scales (SPI, SPEI) most closely correlated with crop yield. The results also suggested that different patterns of
yield response to drought occurred depending on the region, period of the year, and the drought time scale. The differing
responses across the country were related to season and the magnitude of various climate variables.
**Key words:** crop yields, drought, Spain, standardized precipitation index, standardized precipitation evapotranspiration index,
standardized Palmer drought severity index








## 1. Introduction

The Mediterranean region is one of the major areas in Europe likely to be subject to the potential impacts of climate change. Many semiarid regions of southwestern Europe are expected to undergo a critical decline in water availability as a consequence of reduced precipitation and an increase in interannual and intra-annual rainfall variability (IPCC, 2014, EEA, 2017). It is also expected that future changes in the precipitation regime, along with a rise in temperature, will inevitably bring more extreme and severe weather events (Giorgi and Lionello, 2008; Webber et al., 2018; Wigley, 2009) that will impact ecosystems and economic sectors (Asseng et al., 2014; Tack et al., 2015). It has been suggested that precipitation and temperature changes in the western Mediterranean region will lead to more severe and longer drought events in coming decades (Alcamo et al., 2007; Dai, 2011; Forzieri et al., 2016; Giorgi and Lionello, 2008; Spinoni et al., 2018; Vicente-Serrano et al., 2014). This is significant because agriculture plays a key role in food supply; in 2017 it accounted for 2.59% of GDP in Spain, 1.92% in Italy, and 3.53% in Greece (World Bank, 2017).

The agriculture sector is highly vulnerable to drought, as it depends directly on water availability (Hanjra and Qureshi, 2010; Meng et al., 2016; Tsakiris and Tigkas, 2007). Although each crop differs in its resilience to water stress (Liu et al., 2016; Lobell et al., 2011), droughts can cause crop failure if the weather conditions are adverse during the most sensitive stage of crop growth (Lobell and Field, 2007). The adverse impacts of drought have been highlighted in recent severe events, including in 2003 when the agricultural and forestry losses from drought in France, Italy, Germany, Spain, Portugal, and Austria were approximately 13 billion Euros (Fink et al., 2004; García-Herrera et al., 2010). The most recent drought, which mostly affected north–central Europe, caused European farmers to claim agricultural aid because of the low production that resulted (European Commission, 2018).

For these reasons the vulnerability of agricultural production to extreme events, and the quantification of drought impacts on crop yields, have become a focus of interest. In recent years diverse studies in the Mediterranean region have assessed these issues from multiple perspectives. For example, Capa-Morocho et al., (2016) investigated the link between seasonal climate forecasts and crop models in Spain, Loukas and Vasiliades, (2004) used a probabilistic approach to evaluate the spatio-temporal characteristics of drought in an agricultural plain region in Greece, and Moore and Lobell, (2014) estimated the impacts of climate projections on various crop types across Europe.

Droughts are difficult to measure and quantify (Vicente-Serrano et al., 2016), and consequently a wide range of drought indices has been developed to provide tools for quantifying the effects of drought across different sectors (Zargar et al., 2011). In this respect, drought indices are the most widely used method for monitoring drought impacts on agriculture; examples of their use available in the scientific literature include in Europe (Hernandez-Barrera et al., 2016; Potopová et al., 2016a; Sepulcre-Canto et al., 2012; Vergni and Todisco, 2011), America (McEvoy et al., 2012; Quiring and Papakryiakou, 2003) and Asia (Ebrahimpour et al., 2015; Wang et al., 2016a). However, there is no general consensus on the most suitable indices for this purpose (Esfahanian et al., 2017). Despite the existing literature, very few studies (Peña-Gallardo et al., 2018a; Tian et al., 2018) have compared drought indices to identify their appropriateness for monitoring drought impacts on agriculture and for various crop types.

Among Mediterranean countries, agriculture in Spain is particularly sensitive to climate because of the low average precipitation level and its marked interannual variability (Vicente-Serrano, 2006). Spain has been subject to multiple episodes of drought (Domínguez-Castro et al., 2012), with those in the last century being amongst the most severe to have occurred in



Europe (González-Hidalgo et al., 2018; Vicente-Serrano, 2006). In 2017 the agricultural and livestock losses caused by drought
were estimated to be at least 3600 million Euros (UPA, 2017), highlighting the need to establish appropriate tools for
monitoring drought impacts on crops.
Information on crop production is commonly limited in terms of spatial or temporal availability. Recent studies in Spain have
analyzed the impact of drought on various crops since the early 21st century at national or provincial scales (Cantelaube et al.,
2004; Hernandez-Barrera et al., 2016; Páscoa et al., 2016), but few have used yield data at finer resolution (García-León et al.,
2019) . In this study we compared different drought indices using two datasets at different spatial scales: provincial information
provided by the national statistical services, and a regional dataset specifically developed for the study. The objectives of this
study were: (1) to determine the most appropriate and functional drought index among four Palmer-related drought indices
(Palmer drought severity index: PDSI; Palmer hydrological drought index: PHDI; Palmer Z index: Z-index; Palmer modified
drought index: PMDI), and the standardized precipitation evapotranspiration index (SPEI), the standardized precipitation index
(SPI), and the standardized Palmer drought index (SPDI); (2) to identify the temporal response of two main herbaceous rainfed
crops (wheat and barley) to drought; and (3) to determine whether there were common spatial patterns, by comparing the two
datasets at different spatial scales.
**2. Methods and datasets**
**2.1. Crop yield data**
The statistical analysis was conducted using an annual dataset of crop yields for peninsular Spain and the Balearic Islands at
two spatial scales for the two main herbaceous rainfed crops (barley and wheat). We obtained provincial annual yield data
from the National Agricultural Statistics Annuaries published by the Spanish Ministry of Agriculture, Fishing and Environment
(MAPA), available at: https://www.mapa.gob.es/es/estadistica/temas/publicaciones/anuario-de-estadistica/default.aspx (last
accessed: March 2018); these include agricultural statistics since the early 20th century. We used data from 1962 to 2014, to
match climate data that was available for this period. The Gipuzkoa and Vizcaya provinces were not used in the analysis at
the province scale as wheat has not been cultivated there since 1973 and 1989, respectively. We used crop production data
collected by the *Encuesta sobre Superficies y Rendimientos de Cultivos-Survey on surface and crop yields* (Esyrce), an agrarian
yield survey undertaken by the MAPA since 1990. This survey records information about crop production at parcel scale every
year from a sample of parcels. Yield observations were aggregated to the main spatial unit defined for agricultural districts by
the MAPA (Fig. 1). As not all territories were included in this survey until 1993, we only considered the period 1993–2015.
Data on barley production is limited in the Esyrce database, and the agricultural districts considered in this study did not
correspond to all the areas where this crop is cultivated.
For both datasets the unit of measure was the harvested production per unit of harvested area (kg/ha); it did not include any
measure of production related to the area of the crop planted in each province or region. To consider the total area covered by
the crops we used the defined rainfed crop delimited area for Spain, derived from the Corine land cover 2000 database
(http://centrodedescargas.cnig.es/CentroDescargas/catalogo.do?Serie=MPPIF ; last accessed: March 2018).
The spatial resolution of yield data can influence the interpretation of drought impacts on agriculture. Figure 2 shows a
comparison of crop yields for the common period of available information in both datasets (1993–2014). Overall, the average
production was greater at the agricultural district scale than at the provincial scale. Tables S1 and S2 summarize the
relationships between the datasets for each province for the available common period, based on Pearson's correlations
coefficients for wheat and barley yields, respectively. It was surprising that both datasets showed very different temporal





variability in crop yields in the analyzed provinces. Wheat yields showed good agreement and highly significant correlations
between both datasets in provinces including Ávila (r = 0.77), Barcelona (r = 0.69), Burgos (r = 0.82), Cuenca (r = 0.86),
Guadalajara (r = 0.87), León (r = 0.69), Palencia (r = 0.73), Salamanca (r = 0.87), Segovia (r = 0.94), Teruel (r = 0.83),
Valladolid (r = 0.92), and Zamora (r = 0.75), while in other provinces including Castellón, Málaga, Murcia, and Navarra the
correlations were non significant or negative. Thus, the national statistics for these districts were unreliable. For barley yields
the available regional data were more limited, but similar relationships with good agreement and more highly significant
correlations were found among the datasets for the provinces where wheat was also cultivated, including Cáceres (r = 0.48),
Cuenca (r = 0.88), Granada (r = 0.51), Guadalajara (r = 0.86), La Rioja (r = 0.76), and Tarragona (r = 0.88); however, for
Sevilla the correlation was negative and significant (r = −0.35).
Mechanization and innovation in agriculture have increased since last century, resulting in a trend of increased yields (Lobell
and Field, 2007), that is also evident in data for Spain. To remove bias introduced by non-climate factors, and to enable
comparison of yields between the two crop types, the original series were transformed to standardized yield residuals series
(SYR), using the following quadratic polynomial equation:
$$SYRS = \frac{y_{d-\mu}}{\sigma}$$

where $y_d$ is the residuals of the de-trended yield obtained by fitting a linear regression model, μ is the mean of the de-trended
series, and $\sigma$ is the standard deviation of the de-trended yield.
This methodology has been applied in other similar studies (Chen et al., 2016; Potopová et al., 2015; Tian et al., 2018). the
full procedure is described by Lobell and Asner, (2003), Lobell et al., (2011) and Potopová et al., (2015).
**2.2. Climate data**
We used a weekly gridded dataset of meteorological variables (precipitation, maximum and minimum temperature, relative
humidity and sunshine duration) at 1.1 km resolution for peninsular Spain and the Balearic Islands for the period 1962–2015.
The grids were generated from a daily meteorological dataset provided by the Spanish National Meteorological Agency
(AEMET), following quality control and homogenization of the data. Further details on the method and the gridding procedure
are provided by Vicente-Serrano et al., (2017). Reference evapotranspiration (ETo) was calculated using the FAO-56 Penman-
Monteith equation (Allen et al., 1998). Weekly data were aggregated at the monthly scale for calculation of the various drought
indices**.**
**2.3. Methods**
**2.3.1. Drought indices**
**Palmer Drought Severity Indices (PDSIs)**
Palmer (1965) developed the Palmer drought severity index (PDSI). Variations of this index include the Palmer hydrological
drought index (PHDI), the Palmer moisture anomaly index (Z-index), and the Palmer modified drought index (PMDI).
Computation of the Palmer indices (PDSIs) is mainly based on estimation of the ratio between the surface moisture and the
atmospheric humidity demand. Subsequent studies have revealed that spatial comparison among regions is problematic (Alley,
1984; Doesken and Garen, 1991; Heim, 2002). In this context we followed the variation introduced by Wells et al., (2004);
this enables spatial comparison when determining a suitable regional coefficient, developing the self-calibrated PDSIs. PDSIs



are also referred to as uni-scalar indices, which can only be calculated at fixed and unknown timescales (Guttman, 1998;
Vicente-Serrano et al., 2010); this is a limitation of these indices.

### Standardized Precipitation Index (SPI)

The standardized precipitation index (SPI) was introduced by Mckee et al. (1993), and provided a new approach to the
quantification of drought at multiple time scales. The index is based on the conversion of precipitation series to a standard
normal variable having a mean equal to 0 and variance equal to 1, by adjusting an incomplete Gamma distribution. The SPI is
a meteorological index used worldwide, and is especially recommended by The World Meteorological Organization (WMO,
2012) for drought monitoring and early warning.

### Standardized Precipitation Evapotranspiration Index (SPEI)

Vicente-Serrano et al. (2010) proposed the standardized precipitation evapotranspiration index (SPEI) as a drought index that
takes into consideration the effect of atmospheric evaporative demand on drought severity. It provides monthly climate
balances (precipitation minus reference evapotranspiration), and the values are transformed to normal standardized units using
a 3-parameter log-logistic distribution. Following the concept of the SPI, the SPEI enables comparison of drought
characteristics at various time scales among regions, independently of their climatic conditions. The SPEI has been widely
used in drought-related studies, including to investigate the impacts of drought on various crops worldwide (Chen et al., 2016;
Kuhnert et al., 2016; Peña-Gallardo et al., 2018b; Potopová et al., 2016b; Vicente-Serrano et al., 2012).

### Standardized Precipitation Drought Index (SPDI)

The standardized precipitation drought index (SPDI) was developed by Ma et al (2014), and relies on the concept of time
scales. It is considered to be a combined version of the PDSI and the SPEI, because the SPDI accumulates the internal water
valance anomalies (D) obtained in the PDSI scheme at various time scales, and the values are later transformed into z-units
following a standard normal distribution. For this purpose a log-logistic distribution has been used, because this has been
shown to be effective at the global scale (Vicente-Serrano et al., 2015).
The SPEI, SPI, and SPDI are referred to here as multi-scalar indices, and the PDSIs as uni-scalar indices. Thus, the multi-
scalar indices were computed at scales of 1, 12, 18, and 24 months, and along with the PDSIs series were de-trended by
adjusting a linear regression model to enable accurate comparisons with de-trended crop yield information. Following the
same procedure used for the yield series, the residual of each monthly series was summed to the average value for the period.

### 2.3.2.   Correlation between drought indices and crop yields

The relationship between the drought indices and the SYRS for both datasets was assessed by calculating polynomial
correlation coefficients (c) (Baten and Frame, 1959). We used a second-order polynomial regression model, given the common
nonlinear relationship between drought indices and crop production (Páscoa et al., 2016; Zipper et al., 2016). Hereafter, the
references made to correlations refer to results obtained using the polynomial approach. The months of August and September
were excluded from the analysis because they correspond to the post harvest period, and we were considering only the period
from sowing to harvest.
As the month of the year when the greatest correlation between the drought index and the crop yield was not known beforehand,
all 10 monthly series for each index were correlated with the annual yield, and the highest correlation value was used. In the
case of the multi-scalar indices, for each monthly series and time scale we obtained 10 correlations (one for each of the 10



months and the 14 time scales considered in the analysis). Thus, 120 correlations were obtained for each crop and spatial unit
considered in the analysis (only correlations significant at p < 0.05 were considered). In addition, we used the time scale (in
the case of multi-scalar drought indices) and the month in which the strongest correlation was found.
A t-test was performed to assess the significance of the differences in the polynomial regression correlation coefficients
obtained from the drought−yield relationships, to determine whether there were significant similarities or differences among
the indices.
**2.4. Identification of spatial patterns for crop yield response to drought.**
A principal component analysis (PCA) was performed to identify general patterns in the effect of drought on crop yields, in
relation to seasonality of the effects. PCA is a mathematical technique that enables the dimensionality of a large range of
variables to be reduced, by fitting linear combinations of variables. We conducted a T-mode analysis, and used the varimax
method to rotate the components to obtain more spatially robust patterns (Richman, 1986). The monthly series of the monthly
maximum correlation values found from the yield−drought relationship were the variables (one data point per month), and the
provinces and agricultural districts were the cases. We selected two principal components (PC) that in combination explained
> 60% of the variance (individually the other components explained < 5% of the variance), and aggregated each province or
agricultural district according to the maximum loading rule (i.e., assigning each spatial unit to the PC for which the highest
loading value was found). The loadings were expressed in the original correlation magnitudes using the matrix of component
weights.
**3.   Results**
**3.1. Relationship of drought indices to crop yields**
Figure 3 shows the strongest correlation found between the crop yield for each dataset and the monthly drought indices. The
correlations differed substantially between the two groups of indices. Independently of the crop type, month of the year, or the
drought time scale considered, the correlation coefficients for the multi-scalar indices were much higher than those for the uni-
scalar indices. In both cases weaker correlations were found for the wheat crops compared with the barley crops. The PDSI,
PHDI, and PMDI correlations were non significant ($p < 0.05$), but the correlations for the Z-index and the multi-scalar indices
were significant for most provinces and agricultural districts. The correlation values for the three multi-scalar drought indices
were similar. At district scale the average values were $c = 0.57$ and $c = 0.6$ for wheat and barley, respectively, and $c = 0.41$
and $c = 0.48$ at the provincial scale. Thus, the datasets showed a stronger correlation for the drought indices at district scale
than at the provincial scale. In addition, more variability was found in the provincial data than in the regional data, associated
with the length of the available records.
The spatial distribution of the maximum correlation coefficients between the drought indices and the crop yields are shown in
figures 4 and 5, for the province and district scales, respectively. The wheat and barley yield−drought correlations showed a
similar spatial pattern among indices at the province scale. Stronger correlations ($c \geq 0.7$) were found for the SPEI and SPI for
the provinces of Castilla y León (Valladolid, Zamora, Segovia, and Soria), Aragón (Zaragoza and Teruel), Castilla La Mancha
(Guadalajara, Albacete, and Toledo), and the province of Valencia (particularly the cereal agricultural districts). The weakest
correlations were found for the southern (Andalusian) provinces. For the Palmer drought indices, the PMDI and Z-index
showed similar spatial patterns to the multi-scalar indices (especially in the central and northern provinces), but the correlations
were weaker ($c = 0.25−0.6$). For most provinces the weakest correlations were found for the PDSI and PHDI ($c = 0.1−0.25$)
for both crops, with no clear spatial difference in the correlations.





The spatial distribution of correlations between wheat yields and the drought indices at the agricultural district scale showed
clearer patterns than those for the province level. Thus, the response of drought indices at district scale is similar to the response
observed at provincial scale, showing stronger correlations for the multi-scalar indices and weaker correlations for the Palmer
indices, especially the PDSI and PHDI. The distribution of correlations among the multi-scalar indices was very similar. The
most correlated agricultural districts (c ≥ 0.8) were in Castilla y León, especially Valladolid, Segovia, north of Ávila, and
northeast of Salamanca. Similar correlations were found for areas of northeast Spain. There was a gradient in correlations from
north to south, with the exception of some districts in northwestern Málaga, where wheat is extensively cultivated. In addition,
in some districts of Galicia, where expansion of the planted wheat area has not been large, there was a strong relationship
between drought indices and crop yields. The results for barley suggest a similar spatial relationship for the various drought
indices. The highest coefficients were found for the multi-scalar indices, followed by the Z-index and the PMDI, with districts
north of Cáceres, north of Galicia, and in Guadalajara showing correlations in the order of c = 0.8, while the correlations were
weaker (c = 0.25−0.4) in districts in the south of Córdoba and Jaén.

### 3.2. Relationship of drought indices to crop yields: temporal responses

Table 1 summarizes the time scales at which the strongest correlations were found for each of the three multi-scalar indices.
Strongest correlations were found for short time scales (1−3 months) for both datasets and both crops, in general with little
difference between the indices. For wheat, for 52.6% of the agricultural districts the yield was most strongly correlated with
all three drought indices at a time scale of 1−3 months; this was also the case for 49.6% of provinces. In agricultural districts
where wheat is cultivated the strongest correlations were predominantly at the 1-month scale (20.37%), especially for the
SPDI, while for most of the provinces this occurred at the 3-month scale, particular for the SPEI and SPI (23.26%). For barley,
57.4% of the districts and 58.7% of provinces where this crop was grown the strongest correlations were predominantly at 1-
to 3-month time scales. Among the various indices for districts, the SPI showed the strongest correlation at the 1-month scale,
while for provinces the SPEI showed the strongest correlation at the 3-month scale (33.33%).
The multi-scalar drought indices showed similar results. Among these, the SPEI was the index most strongly correlated with
yield in the highest percentage of provinces and districts (Table 2). For wheat crops the SPEI was the most strongly correlated
index with yield in ~37% of the agricultural districts and ~58% of the provinces; these correlations were found predominantly
at the 3-month time scale. For this crop the SPDI was most strongly correlated with yield in a similar proportion of districts
(~33%), primarily at the 1-month scale, but only ~14% at the province scale. In general, most of the maximum correlations
corresponded to short time scales.
Figure 6 shows the spatial distribution of the most strongly correlated drought indices. For most of the provinces the SPEI was
the index most strongly correlated with crop yield. For the agricultural districts there was substantial spatial variability and,
along with the provincial results, no well-defined spatial pattern that distinguished specific areas for which one index was most
effective at monitoring drought. For barley the SPDI showed the best correlation with yield among districts (~44%), while in
provinces the SPEI was best correlated (~69%). No clear spatial patterns were evident. The similarities in the magnitude of
the correlations between multi-scalar drought indices and crop yields were statistically significant. A t-test (Fig. S1) was used
to determine whether there were significant differences in the magnitude of correlations obtained using the various multi-
scalar drought indices. This showed significant differences between the SPEI and the SPDI in ~30% of agricultural districts
where wheat was grown; these were districts that showed a weaker correlation of yield with drought indices. The results
suggest that, for districts having strong correlations between drought indices and crop yields, the two indexes were equally
useful. A lower proportion of districts where barley is planted showed that statistical differences among indices exist. In





263 contrast, for provinces no significant differences were found. Overall, this suggests the appropriateness of using any of these

264 multi-scalar indices indistinctly.


266 **3.3. Spatial patterns of drought index correlations at the monthly scale**

267 Regionalization of the crop yield response to drought based on monthly correlations with the drought indices was undertaken

268 in relation to the most correlated drought index in each region, independently of the month in which this maximum correlation

269 occurred. Thus, in this analysis the results obtained using the various multi-scalar drought indices were merged. General spatial

270 patterns in the effect of drought conditions on yield were identified using a T-mode PCA. Figures 7 and 8 show the results for

271 the provincial and regional datasets, respectively. We selected two components that explained more than the 60% of the

272 variance in each case. This classification reinforced the north−south pattern of correlations previously found for both datasets.

273 Figure 9 shows the time scales for which the maximum monthly correlations were found for the provinces and agricultural

274 districts for each of the defined components, using a maximum loading rule.

275 **3.3.1. Wheat**

276 *Agricultural district scale*

277  At the district scale the PCA for wheat (Figure 7a) showed more defined spatial patterns than did the PCA at the

278 provincial scale. PC1 explained 43.36% of the variance, and was characterized by stronger correlations ($c = 0.7$−$0.9$) in districts

279 mainly located on the north and central plateau; these were stronger than those recorded for the same locations at the provincial

280 scale. Weaker correlations ($c = 0.15$−$0.5$) were dispersed, although these were found predominantly in the south and northwest.

281 The scores for PC1 showed particular sensitivity to drought during spring, although strong correlations were also found during

282 autumn. PC2 explained 18.63% of the variance, and the loading coefficients also showed a clear spatial pattern, with the

283 agricultural districts north of Sevilla and east of Castilla La Mancha having the highest values. The weakest correlations were

284 found for the districts of Andalucía, Extremadura, and Aragón. Lower scores in PC2 characterized the interannual response to

285 drought relative to PC1. These districts in PC2 also showed a stronger response during spring but not autumn, as was found

286 for PC1. The distribution of PCs according to the maximum loading rule enabled identification of a north−south component

287 in the sensitivity of wheat yields to the drought index. The time scales at which wheat yields in agricultural districts responded

288 most during spring varied from shorter time scales (3-month) in districts in PC1 to longer time scales (5- to 6-month) for those

289 in PC2 (Fig. 9e, 9f), which also showed greater variability in most months relative to districts from PC1. Greater variability

290 for wheat at the district scale was observed relative to that at the provincial scale. Due to the major number of observations

291 considered, the response to drought in Spain when considering district scale shows more heterogeneity than at provincial scale.

292 *Provincial scale*

293 The results for wheat at the provincial scale (Fig. 7b) showed that the first (PC1) and second (PC2) components explained

294 51.7% and 20.8% of the variance, respectively. The loadings of the first component were higher for the central plateau and the

295 east of Spain. These represent provinces in the Castilla y León and Castilla y La Mancha districts, and the provinces of

296 Castellón, Valencia, Alicante, Cantabria and Huelva, and Sevilla and Almería in Andalucía. In these provinces there was a

297 strong correlation between drought indices and crop yields, especially during spring, with particularly strong correlations in

298 May. In contrast, during winter the correlations were weaker, especially in February. PC2 showed greater spatial heterogeneity,

299 with strong correlations in the east (Zaragoza and Tarragona provinces) and south (Cádiz, Córdoba, Málaga, Granada, and

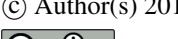


Jaén provinces) of Spain. For this component the temporal response to drought was not as strong as that for PC1, but the
maximum correlation was also found during May. The distribution of the maximum loadings showed a dispersed pattern, with
PC1 grouping provinces in the central plateau and east of Spain, and PC2 grouping those in southern and some northeastern
provinces. The averaged temporal response to drought during spring is set at medium time scales (4−7 months). In particular,
in May most of the provinces correlated at 5 months (Fig. 9a, 9b), indicating the importance of climatic conditions during
winter and spring to the crop yields obtained. This was also evident for the longer time scales at which most of the provinces
correlated during the winter months (11−18 months). It is noteworthy that there was great variability in the temporal response
of provinces in PC1 in October, February, March, and April.

### 3.3.2.  Barley

*Agricultural* district *scale*

For barley crops (Fig. 8a) both components showed strong correlations (c = 0.6−0.9) in most of the agricultural districts. In
general, the districts showing the strongest correlations in PC1 and PC2 were those located in Castilla La Mancha, and north
of Cáceres and Córdoba. Scores for PC1 for barley crops were similar to those for PC1 for wheat during spring and autumn,
but the results for PC2 suggest that there was little interannual sensitivity to drought. Most of the correlations for spring indicate
that barley responded to drought conditions at the 3−4 month scale, mainly in those districts associated with PC1. Barley yields
in districts associated with PC2 were more affected by drought conditions in May at 7−9 month time scales (Fig. 9g, 9h).

*Provincial scale*

For barley at the provincial scale (Fig. 8b) we found more variability in the magnitude of correlations. For PC1 (explaining
43.22% of the variance) strong correlations (r = 0.7−0.9) were found for the north and central provinces of Castilla y León,
the central provinces of Castilla y la Mancha, and Madrid, Teruel, Valencia and Castellón. The provinces associated with PC2
(explaining 27.91% of the variance) were more dispersed than those in PC1, and those showing show strong correlations
included Zaragoza and Guadalajara in the north, Barcelona and Balearic Islands in the northeast and east, Cáceres in the west,
and Cádiz, Córdoba, Málaga, Granada and Jaén in the south. Provinces showing weaker correlations in PC1 were spread in
the northeast (e.g., Navarra, Zaragoza, and Lleida) and west of Spain (e.g., Cáceres and Badajoz). Component scores for PC1
were higher than for PC2, although for wheat crops both showed maximum scores during spring (March) and minimum scores
in autumn and winter. More provinces in May were correlated with drought indices at medium drought time scales (4−8
months). During spring, provinces in PC1 showed correlations at longer time scales (7−8 months), while provinces in PC2
showed responses at shorter time scales (3−4 months) (Fig. 9c, 9d).

### 3.3.3.  General    climatological    characteristics    for    the    PCA    components

Figures S2-11 show the distribution of climatic characteristics including precipitation, atmospheric evaporative demand
(AED), maximum and minimum temperature, and the hydroclimatic balance (precipitation minus AED) at the district scale
for the two PCA components. For those districts where wheat was cultivated, no major differences in AED values were found
among the components. However, minor differences were observed in precipitation among districts belonging to different
PCA components. Those in PC2 had on average less precipitation than those in PC1, especially during autumn, but the
difference was not substantial. Greater differences were observed for temperature, with PC1 mainly characterized by districts
that had higher maximum temperatures in autumn and spring, and with higher minimum temperatures than the districts in PC2.
These results highlight the important role of temperature in the different responses of crop yield to drought, and demonstrate





that, contrary to what may have been expected, temperature and not precipitation was the main factor constraining crop growth.
Thus, changes in extreme temperature levels may influence future crop yields. Districts in PC2 where the barley yield
correlated with drought indices were characterized by lower levels of precipitation and higher maximum and minimum
temperatures than districts represented by PC1, and by higher AED, especially from April to July. Extremes of temperature
also seemed to be the major factor determining barley crop yield.

### 4. Discussion

In this study we investigated the impacts of drought on two rainfed crops in Spain, as measured by a variety of drought indices.
We used two datasets of annual crop yields, one from agricultural statistics at the provincial scale spanning the period
1962–2013, and the other a new database at the agricultural district scale from the available parcel data from the national
survey covering the period 1993–2015. To identify the best indicator of the impact of drought on yields and their sensitivity
to climate, we evaluated the performance of seven drought indices. The selection of drought indices was based on those
commonly used to monitoring droughts worldwide, including the standardized precipitation and evapotranspiration index
(SPEI), the standardized precipitation index (SPI), the Palmer drought indices (PDSI, Z-Index, PHDI, and PMDI), and the
standardized Palmer drought index (SPDI).
Independently of the type of crop and the temporal scale considered, our results showed that drought indices calculated at
different time scales (the SPEI, the SPI, and the SPDI) had greater capacity to reflect the impacts of climate on crop yields,
relative to uni-scalar drought indices. The better performance of these multi-scalar drought indices was mainly because of their
flexibility in reflecting the negative impacts of drought over a range of regions having very different characteristics (Vicente-
Serrano et al., 2011). This issue is especially relevant in agriculture, as vegetation components do not respond equally to water
deficit. The sensitivity and vulnerability of each type of crop to drought, and the characteristics of the specific region influence
the variability evident in the response to droughts (Contreras and Hunink, 2015). Nonetheless, the results of the assessment of
the performance of the PDSIs demonstrated that correlations varied markedly among them, showing some exceptions that may
affect their usefulness for monitoring purposes. Overall, our results showed that the PHDI had the weakest relationship to crop
yields, followed by the PDSI and the PMDI. The better performance of the PDSI over the PHDI was expected, as the latter
was primarily developed for hydrological purposes. Likewise, our results confirmed a better performance of the PMDI (a
modified version of the PDSI) over the original PDSI for both crops. Our results are consistent with those of previous studies
assessing agricultural drought impacts on crop yields at the global (Vicente-Serrano et al., 2012) and regional (Peña-Gallardo
et al., 2018b) scales. The Z-index was the best uni-scalar index among the set analyzed in our study. This index measures
short-term moisture conditions, which is a major factor in crop stress (Quiring and Papakryiakou, 2003). Thus, the Z-index
was more closely correlated with crop yield than any of the other Palmer indices, indicating its usefulness relative to other
PDSIs (Karl, 1986).
Although our findings point to poorer performance of the Palmer drought indices relative to the multi-scalar drought indices,
they remain among the most widely accepted indices. Numerous studies have used the Palmer indices in assessments of the
use of drought indices for monitoring agricultural drought in various regions worldwide, and have reported the superiority of
the Z-index (Mavromatis, 2007; Quiring and Papakryiakou, 2003; Sun et al., 2012; Tunalıoğlu and Durdu, 2012) ; our results
confirm it usefulness among the Palmer drought indices.
Nevertheless, it is important to stress that the usefulness of PDSIs is less than drought indices that can be computed at different
time scales (Vicente-Serrano et al. 2012). We demonstrated that the three multi-scalar drought indices in our study (SPEI, SPI,
and SPDI) were able to detect drought at different time scales, enabling past weather conditions to be related to present



conditions in regions characterized by diverse climatic conditions. This is consistent with previous comparative studies in
various regions that reported multi-scalar drought indices were effective for monitoring drought impacts on agricultural lands
(Blanc, 2012; Kim et al., 2012; Potopová, 2011; Potopová et al., 2016a; Tian et al., 2018; Zhu et al., 2016; Zipper et al., 2016).
Although previous studies reported differences among some of the above three indices (e.g., the SPDI and the SPEI; Ghabaei
Sough et al., 2018), others have reported similarities in their performance in assessing agricultural drought impacts (Labudová
et al., 2016; Peña-Gallardo et al., 2018a). The similar magnitudes of their correlations suggest a similar ability to characterize
the impact of drought on crop yields. However, minor differences among these indices suggested the SPEI performed best.
First, for both crops slightly stronger correlations were observed with the SPEI, although the SPDI was superior in relation to
barley yields at the agricultural district scale. In general, the SPEI was found to be the most suitable drought index in the
majority of agricultural districts and provinces. This suggests that inclusion of AED in the drought index calculation, as occurs
in the SPEI, provides greater capacity to predict drought impacts on crop yields compared with the use of precipitation only.
Variation in the maximum and minimum temperatures has been found to be the major factor differentiating agricultural
districts and provinces having greater sensitivity to drought. Previous studies have stressed the risks associated with an increase
in global temperatures, particularly maximum temperatures, and the possible effects on crop yields (Lobell and Field, 2007;
Moore and Lobell, 2014). Thus, a ~5.4% reduction in grain yields resulting from an increase in average temperature is expected
to occur under the current global warming scenario (Asseng et al., 2014; Zhao et al., 2017).
The temporal and spatial effects of drought on yields seem to be very complex, given the observed variability in Spain. In this
respect, significant yield effects of drought were found in both datasets. Nevertheless, at the agricultural district scale there
was a more evident spatial effect of drought on agricultural yields. This is a key finding for spatial-scale analyses, although
the lack of long time series datasets on regional yields is a common constraint.
Drought effects on barley and wheat were similar in space and time, although their sensitivity to drought differed, as shown
by differences in the magnitude of the correlations with the drought indices, with wheat yields showing stronger correlations
than barley yields. This can be explained by the different physiological characteristics of the two crops, as barley is less
dependent on water availability at germination and the grain filling stage than wheat (Mamnouie et al., 2006). Although the
transpiration coefficient for barley is higher, this crop is not as subject as wheat to water stress under drought conditions
(Fischer et al., 1998). Our results indicate that the temporal responses of barley and wheat to drought conditions were very
similar, despite the fact that in Spain barley is typically cultivated later than wheat, and in soils having poor moisture retention.
Therefore, the phenological characteristics of each type of crop determine how drought affects yields. The results showed that
temperature had a more important role than precipitation, suggesting that extreme variations in average temperature conditions
during the most sensitive growth stages may have a negative impact on crops.
Overall, crop yields in Spain tend to respond to short drought time scales (1−3 months). However, the sensitivity of crops to
drought is greater during spring at medium (4−6 months) time scales. This highlights that moisture conditions during winter
(the period corresponding to planting, and the first growth stages of tillering and stem elongation), are crucial for the successful
development of the plants (Çakir, 2004; Moorhead et al., 2015; Wang et al., 2016a, 2016b).
We found a stronger response of crops to climatic conditions in provinces and agricultural districts in the central plateau, and
unexpectedly a weaker response in southwestern districts. This reflects the inconsistencies reported for the Iberian Peninsula
by Páscoa et al., (2016) , who argued that spatial differences can be explained mainly by the differing productivities in the
various districts; we noted this for the mainly agrarian areas of peninsular Spain (Castilla y León and Castilla La Mancha),
and the characteristically heterogeneity of this territory. In the southwestern agricultural areas, where the precipitation rates





are lower and temperatures higher, the correlations of yield with drought were weaker. This can be attributed to episodes of
abnormal extreme temperatures, such as the very low temperatures in early spring or warmer than usual temperatures in winter;
these would affect the expected low evapotranspiration rates during the cold season (Fontana et al., 2015; Kolář et al., 2014).
A recent study by Hernandez-Barrera et al., (2016) demonstrated that during autumn and spring, precipitation deficit is the
most influential climate factor affecting wheat growth, while an increase in the diurnal temperature range causes a reduction
in wheat yield. We found no major differences in precipitation among districts belonging to any of the two defined components,
but found other differences including in the average maximum and minimum temperatures. These findings highlight the
complexity in choosing a useful drought index that encompasses the specificities of each crop, including its sensitivity to
moisture and environmental conditions throughout the entire growth cycle, and its seasonality. This underscores the importance
of testing and comparing the appropriateness of different drought indices to ensure accurate identification of the multi-temporal
impacts of drought on natural systems.

### 5.  Conclusions

The main findings of this study are summarized below.
(1)  Assessment of the efficacy of drought indices for monitoring the effect of climate on agricultural yields demonstrated
the better performance of multi-scalar indices. The ability to calculate these indices at various time scales enabled
drought impacts to be more precisely defined than with the use of indices lacking this characteristic. The multi-scalar
drought indices assessed also had fewer computational and data requirements (particularly the SPEI and the SPI),
which is a significant consideration when performing analyses based on scarce climate data.
(2)  From a quantitative evaluation of the relationship of drought indices to crop yields we determined that both of the
multi-scalar drought indices tested were useful for assessment of agricultural drought in Spain. However, the SPEI
had slightly better correlations and is the most highly recommended for the purpose.
(3)  The spatial definition of yield responses to drought was clearer at the district scale, where the finer spatial resolution
enabled better definition of the patterns of responses because the climatic variability of each region was better
captured at this scale.
(4)  Barley and wheat yields were more vulnerable to drought during spring, both at short (1−3 months) and medium (4−6
months) time scales. Moisture conditions during late autumn and winter also had an impact on the crop yields.
(5)  The strongest relationships between drought indices and crop yields were found for the northern and central
agricultural districts. The relationships for the southern districts were weaker because of the difficulty of
characterizing drought impacts over the diverse and complex territory involved.
(6)  The climatic and agricultural conditions in Spain are very diverse. The large spatial diversity and complexity of
droughts highlights the need to establish accurate and effective indices to monitor the variable evolution of drought
in vulnerable agriculture areas. Climate change is likely to lead to yield losses because of increased drought stress on
crops, so in this context effective monitoring tools are of utmost importance. The authors consider that further analysis
complementing this study may help to unravel the climate mechanisms that influence the spatio-temporal responses
of yields to climate in Spain.








**Acknowledgments**

This work was supported by the research projects PCIN-2015-220 and CGL2014-52135-C03-01 financed by the Spanish Commission of Science and Technology and FEDER, IMDROFLOOD financed by the Water Works 2014 co-funded call of the European Commission and INDECIS, which is part of ERA4CS, and ERA-NET initiated by JPI Climate, and funded by FORMAS (SE), DLR (DE), BMWFW (AT), IFD (DK), MINECO (ES), ANR (FR) with co-funding by the European Union (Grant 690462). Peña-Gallardo Marina was granted by the Spanish Ministry of Economy and Competitiveness (BES-2015-072022).

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




**Tables**

Table 1. Percentage of analyzed agricultural districts and provinces where wheat and barley are cultivated, at which the maximum correlations per time scale were found using the multi-scalar indices.

| Time-scale | | 1 | 2 | 3 | 4 | 5 | 6 | 7 | 8 | 9 | 10 | 11 | 12 | 18 | 24 |
|---|---|---|---|---|---|---|---|---|---|---|---|---|---|---|---|
| | | | | | | a) | **Agricultural district data** | | | | | | | | |
| Wheat | SPI | 18.38 | 15.38 | 13.68 | 9.83 | 4.27 | 7.26 | 2.56 | 5.13 | 1.28 | 3.42 | 6.41 | 2.14 | 5.98 | 4.27 |
| | SPEI | 16.67 | 14.96 | 17.09 | 9.83 | 6.41 | 3.42 | 5.13 | 4.7 | 3.42 | 2.56 | 3.85 | 4.27 | 5.13 | 2.56 |
| | SPDI | 26.07 | 21.79 | 13.68 | 5.13 | 3.42 | 2.99 | 2.56 | 2.56 | 2.14 | 5.13 | 1.71 | 3.85 | 3.42 | 5.56 |
| **Averaged %** | | **20.37** | **17.38** | **14.82** | **8.26** | **4.70** | **4.56** | **3.42** | **4.13** | **2.28** | **3.70** | **3.99** | **3.42** | **4.84** | **4.13** |
| Barley | SPI | 29.63 | 14.81 | 14.81 | 12.96 | 0 | 3.7 | 3.7 | 1.85 | 3.7 | 1.85 | 1.85 | 3.7 | 3.7 | 3.7 |
| | SPEI | 24.07 | 12.96 | 22.22 | 9.26 | 1.85 | 3.7 | 5.56 | 3.7 | 3.7 | 1.85 | 0 | 5.56 | 1.85 | 3.7 |
| | SPDI | 24.07 | 14.81 | 14.81 | 7.41 | 7.41 | 3.7 | 11.11 | 1.85 | 0 | 3.7 | 0 | 0 | 3.7 | 7.41 |
| **Averaged %** | | **25.92** | **14.19** | **17.28** | **9.88** | **3.09** | **3.70** | **6.79** | **2.47** | **2.47** | **2.47** | **0.62** | **3.09** | **3.08** | **4.94** |
| | | | | | | b) | **Provincial data** | | | | | | | | |
| Wheat | SPI | 6.98 | 13.95 | 23.26 | 6.98 | 2.33 | 6.98 | 6.98 | 6.98 | 6.98 | 2.33 | 4.65 | 4.65 | 4.65 | 2.33 |

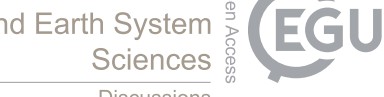

|  |  | | | | | | | | | | | | | |
|---|---|---|---|---|---|---|---|---|---|---|---|---|---|---|
| SPEI | 9.3 | 11.63 | 23.26 | 11.63 | 9.3 | 0 | 6.98 | 6.98 | 2.33 | 2.33 | 4.65 | 4.65 | 4.65 | 2.33 |
| SPDI | 13.95 | 32.56 | 13.95 | 2.33 | 4.65 | 4.65 | 6.98 | 0 | 2.33 | 6.98 | 6.98 | 2.33 | 0 | 6.98 |
| **Averaged %** | **10.08** | **19.38** | **20.16** | **6.98** | **4.65** | **3.88** | **6.20** | **6.98** | **3.10** | **2.33** | **5.43** | **3.88** | **3.10** | **3.88** |
| SPI | 7.14 | 19.05 | 30.95 | 9.52 | 4.76 | 7.14 | 0 | 2.38 | 0 | 0 | 0 | 11.9 | 0 | 4.76 |
| SPEI (Barley) | 11.9 | 11.9 | 33.33 | 7.14 | 4.76 | 4.76 | 7.14 | 4.76 | 7.14 | 0 | 0 | 2.38 | 2.38 | 2.38 |
| SPDI | 9.52 | 38.1 | 14.29 | 4.76 | 4.76 | 0 | 0 | 0 | 0 | 2.38 | 2.38 | 4.76 | 2.38 | 4.76 |
| **Averaged %** | **9.52** | **23.02** | **26.19** | **7.14** | **4.76** | **6.35** | **2.38** | **2.38** | **5.55** | **0.00** | **0.79** | **6.35** | **1.59** | **3.97** |





Table 2. Percentage of analyzed agricultural districts and provinces where wheat and barley are cultivated, where the maximum correlations with the multi-scalar indices
were found. Information in parentheses show the time scale at which the provinces and agricultural districts correlate most and the percentage of the provinces and
district.

|  |  | SPEI | SPDI | SPI |
|---|---|---|---|---|
| **Agricultural districts** | **Wheat** | 36.75 (3, 7.26) | 33.33 (1, 7.69) | 29.91 (2, 4.70) |
|  | **Barley** | 35.19 (3, 11.11) | 44.44 (1, 12.96) | 20.37 (1, 11.11) |
| **Provinces** | **Wheat** | 58.14 (3, 18.60) | 13.95 (24, 4.65) | 27.9 (3, 4.65) |
|  | **Barley** | 69.04 (3, 16.66) | 9.52 (1, 7.14) | 21.42 (5,24, 4.76) |






**Figures**

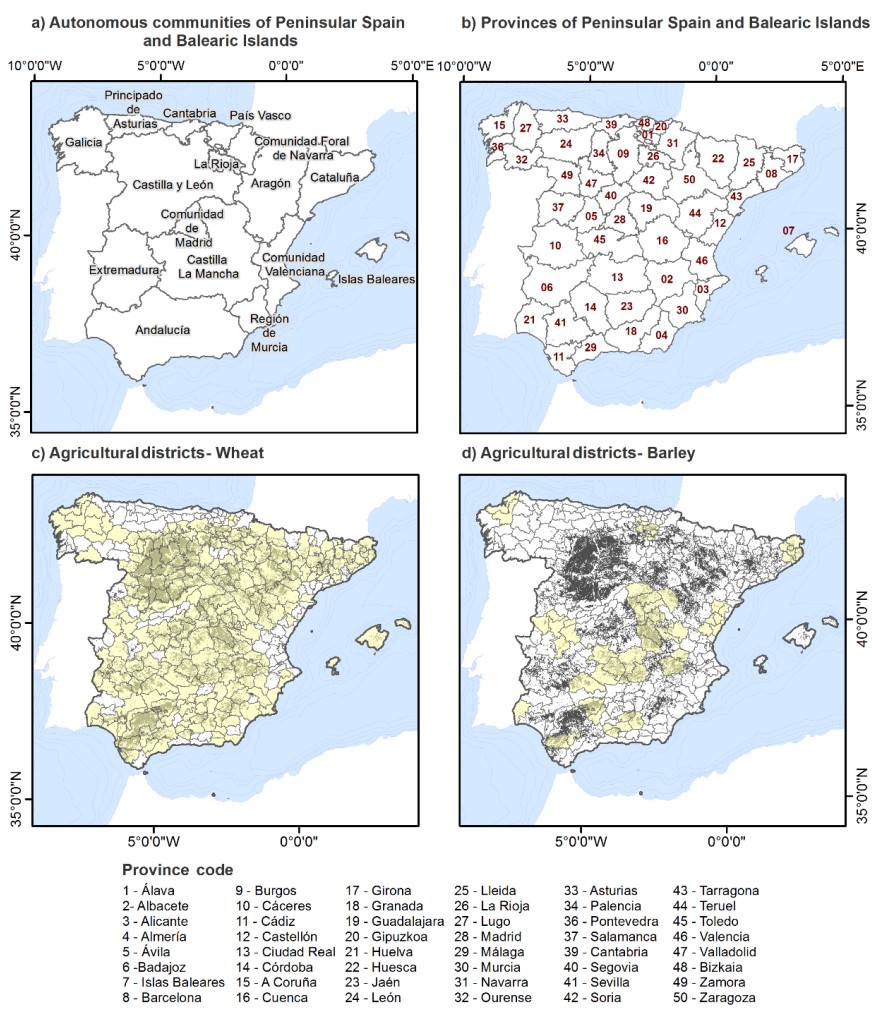


Fig. 1. Location of Spanish Autonomous Communities (a) and provinces (b), and the distribution of
agricultural districts having data available (yellow) for wheat (c) and barley (d) yields for the period
1993–2015. Areas where rainfed cereal crops are cultivated (Corine Land Cover 2006) are shown in grey.





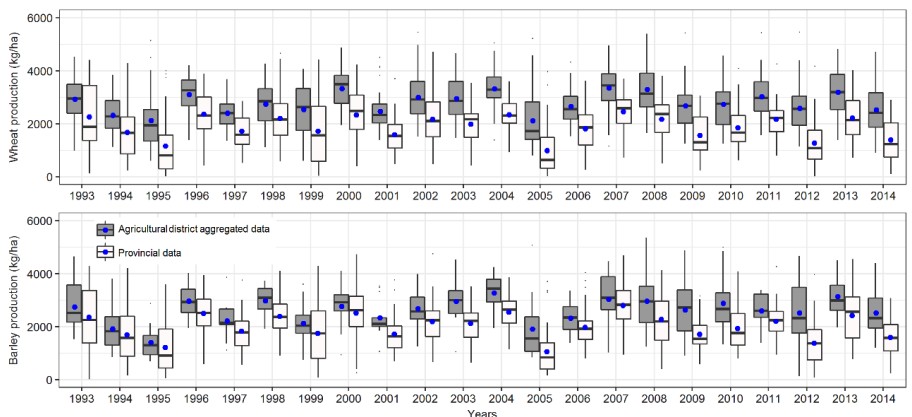


Fig. 2. Temporal series of wheat (top) and barley (bottom) yields for the provincial data, and the aggregated
agricultural district data at the province scale for the common period 1993–2014. The solid black line shows
the median and the blue dot shows the mean.

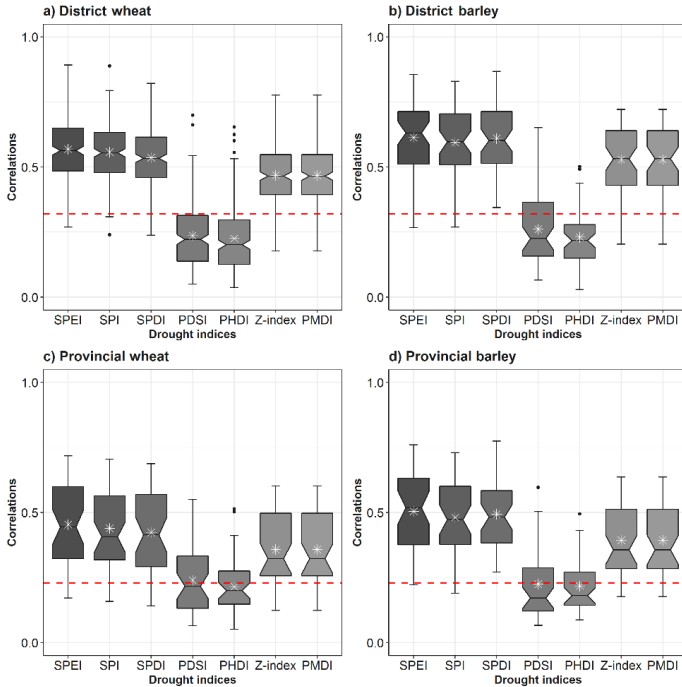


Fig. 3. Box plots showing the strongest correlation coefficients found between drought indices and wheat
and barley yields at the agricultural district (a and b) and provincial (c and d) scales, respectively. The solid
black line shows the median, the white asterisk shows the mean, and the dashed red lines show the $p < 0.05$
significance level.



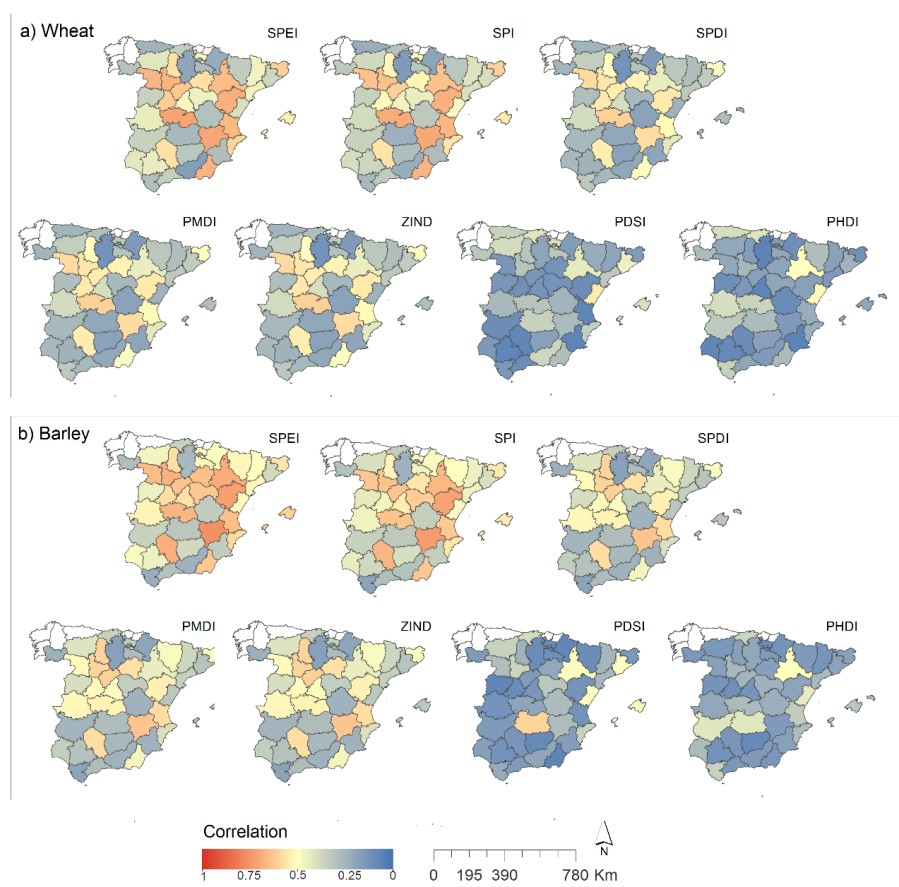


Fig. 4. Spatial distribution of the highest correlation coefficients between the drought indices and the wheat

(a) and barley (b) yields at the provincial scale, independently of the time scale.






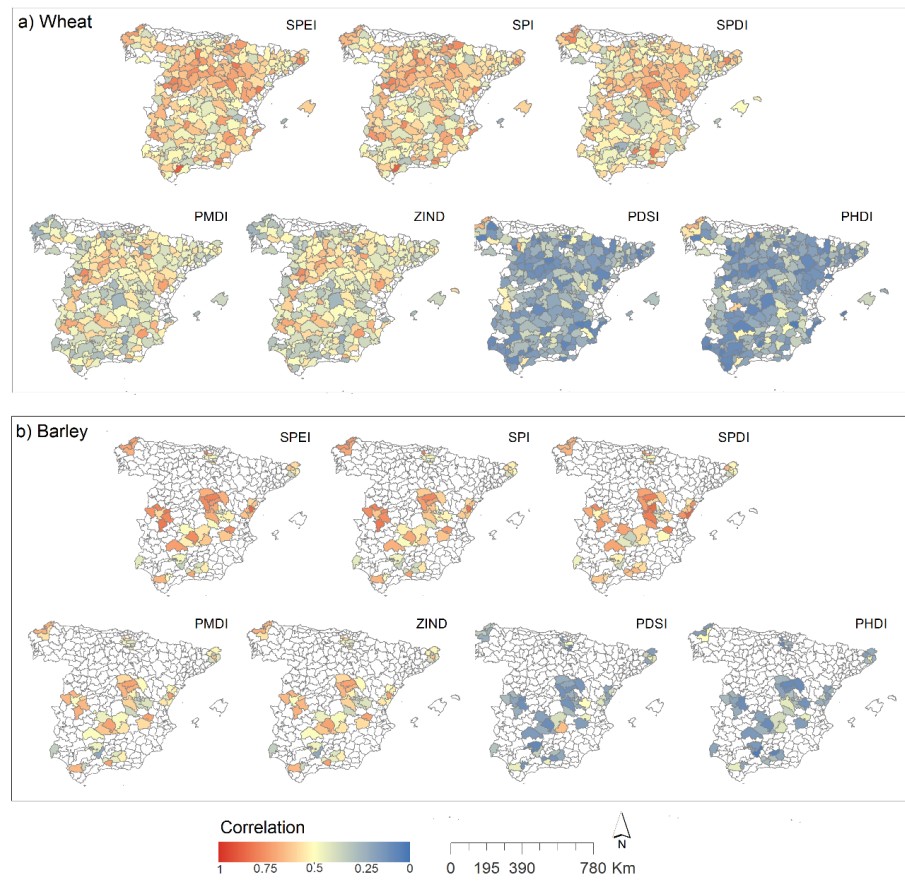


Fig. 5. Spatial distribution of the highest correlation coefficients between the drought indices and the wheat
(a) and barley (b) yields at the agricultural district scale, independently of the time scale.

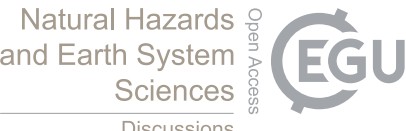



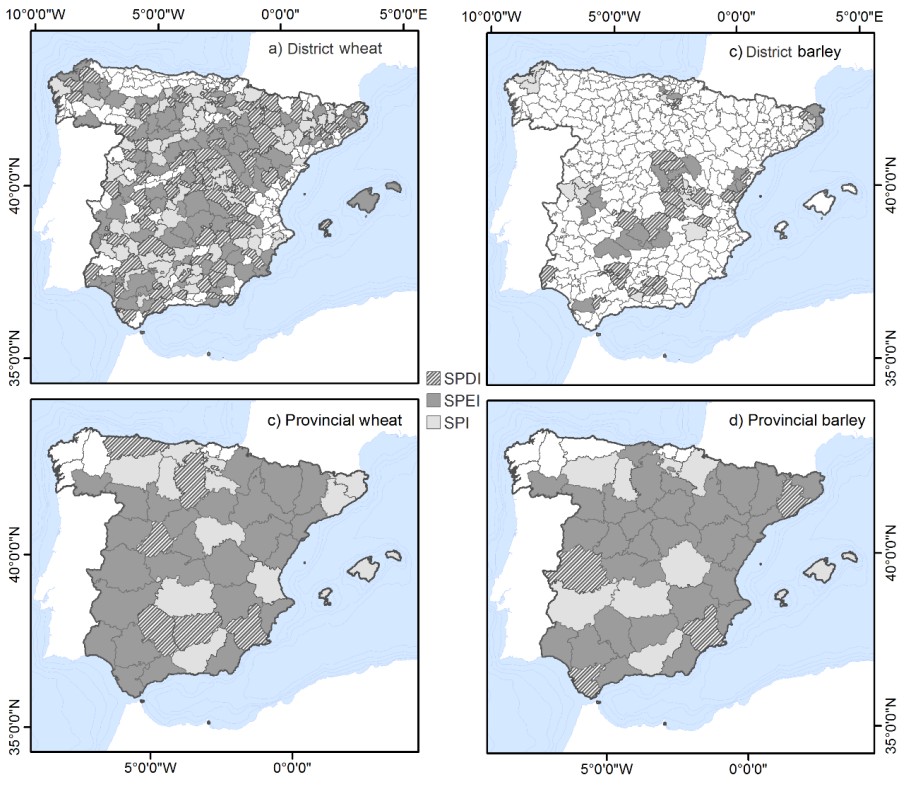

Fig. 6. Spatial distribution of the drought indices having the strongest correlations with wheat (left) and barley (right) at the province (bottom) and agricultural district (top) scales.





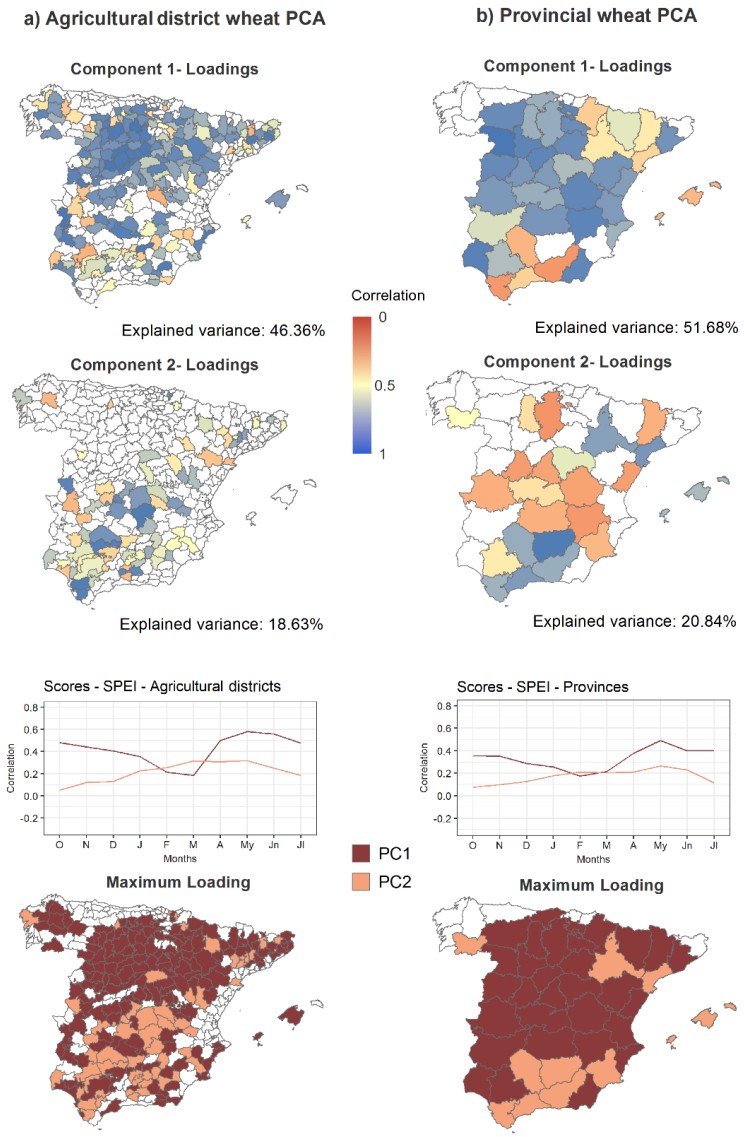

744

Fig. 7. PC loadings, PC scores, time scales, and maximum loading rules from the PCA for monthly

maximum correlation coefficients between the SPEI and wheat yields at the agricultural district (a) and

provincial (b) scales, independently of the time scale. The PC loadings and maximum loadings were

significant at $p < 0.05$.





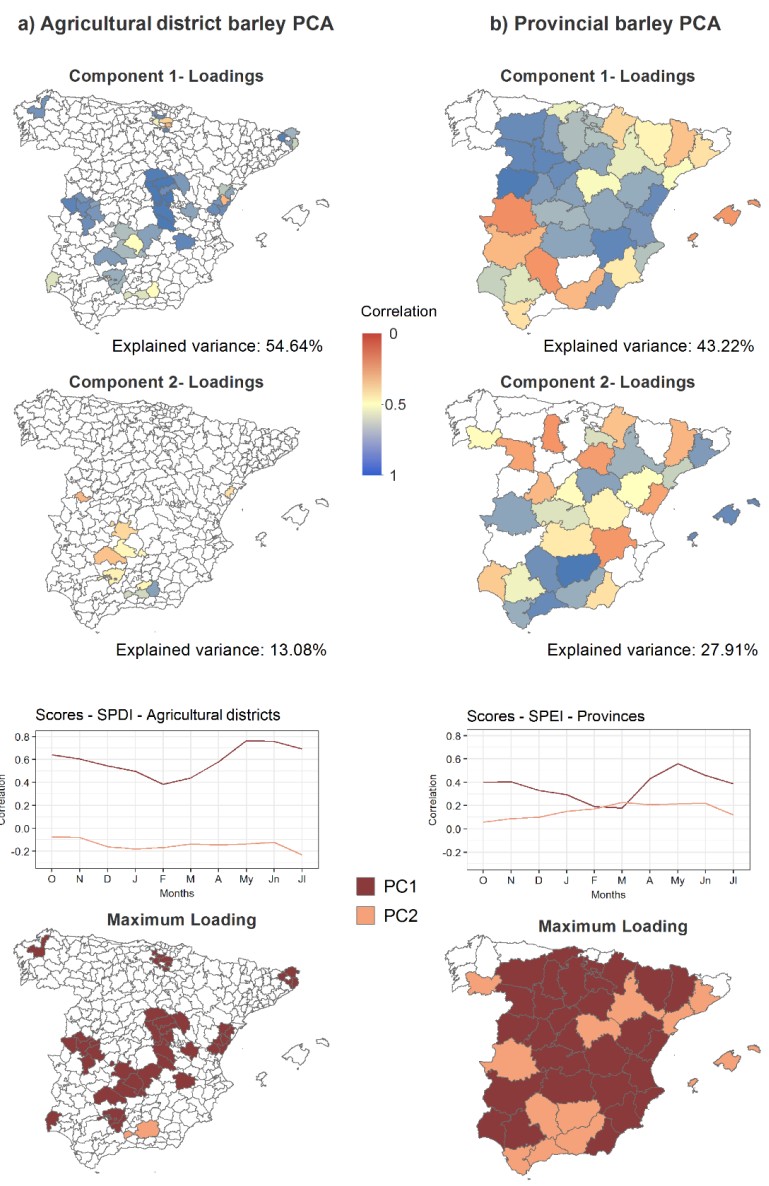

749

Fig 8. PC loadings, PC scores, time scales, and maximum loading rules from the PCA for monthly
maximum correlation coefficients between the SPEI and barley yields at the agricultural district scale (a),
and the SPDI and barley yields at the provincial scale (b), independently of the time scale. The PC loadings
and maximum loadings were significant at $p < 0.05$.






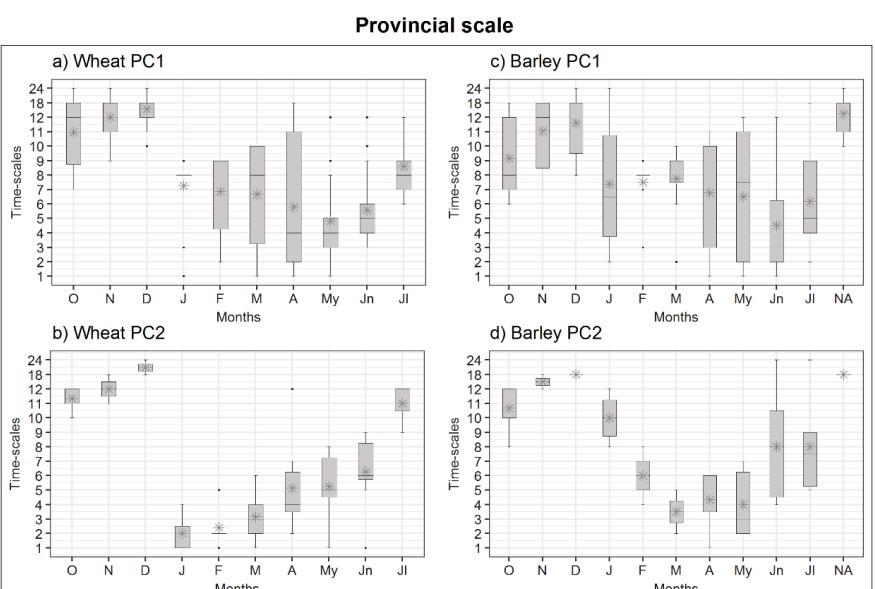

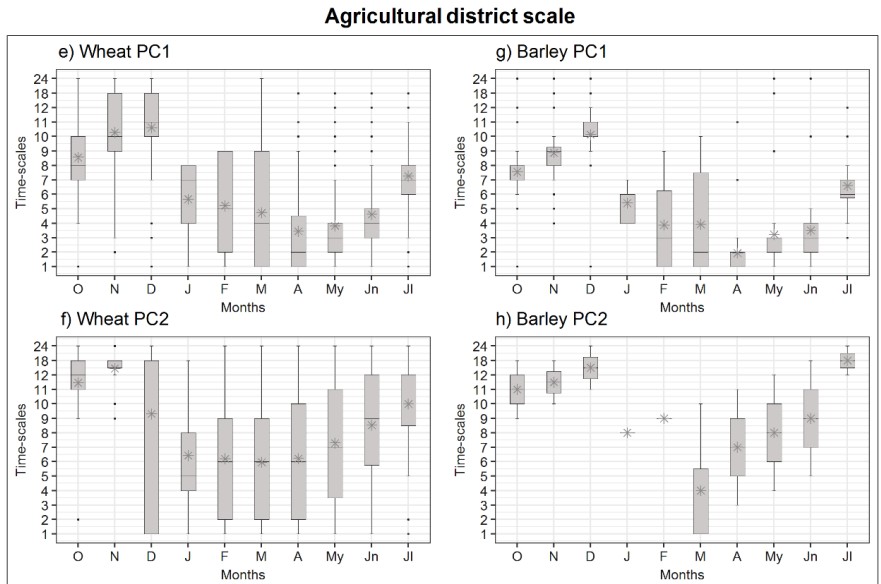


Fig. 9. Box plots showing the time scale at which significant monthly correlations were found in the
provinces (top) and agricultural districts (bottom) for wheat and barley for each of the components defined
in the PCA.