# Peer review of "The impact of drought on the productivity of two rainfed crops in Spain"

_Natural Hazards and Earth System Sciences, 2019_

## Referee Comment (RC1) · Anonymous Referee #1 · 8 Feb 2019

This article is interesting by combining data from different drought indices and yield of two crops at district level in region with water insufficiency. The methods used for statistical analysis of data sets are reasonable and provide an original contribution to the regional crop-loss and drought assessment in Spain. I have a few comments that need clarification before the paper can be considered for publication:

- Line 127: Please re-write SYR as SYRS. Morover, the short name of the SYRS (standardised yield residuals series) was firstly introduced by Potopova et al. 2015. - Please include in sec 2.1 as example plots with temporal trends of wheat and barley (t/ha) and plots with temporal evolution of SYRS of wheat and barley during the study period. - Line 177: The citation of polynomial correlation coeff. not be icluded, it is well known. - References: Please carefully check your references. Some reference

are having the wrong names throughout the text and the References section, For ex. Potop, 2011.

---

## Referee Comment (RC2) · Anonymous Referee #2 · 22 Feb 2019

This manuscript examines the spatial and temporal relationships of crop yield to a suite of drought indices in Spain. Two different crop yield datasets are used and seven drought indices are used including the multi-scalar SPEI, SPI, and SPDI and the uni-scalar PDSI, z-index, PHDI, and PMDI. Main findings were that then multi-scalar drought indices were most strongly related to crop yield and that spatial patterns varied depending on time of year and drought index time scale.

This paper is well written and of interest to the community. Overall, I did not find any major issues with the paper, but a few parts of the methods and analysis do need a bit more clarification. Once these methods are clarified the paper should be accepted for publication.

Comments:

[Figure]

1. Figures 3-5 show the strongest correlations to crop yield but don't discuss the ending month of the strongest correlations. I realize month is discussed in other figures but I don't think it specifically highlights the ending month of those maximum correlations. Please clarify how the ending month is accounted for. An additional figure showing the spatial distribution of the ending months associated with max correlations could be useful.

2. Throughout the paper the "impact of drought" on crops is mentioned several times. However, all data for the period is used which includes droughts, wet years, and average years. I would think additional analysis would be needed to justify that wording that looks specifically at drought episodes. I would suggest somehow rephrasing that throughout the manuscript to more of a general "climate" impact of droughts as drought events are never discussed.

3. It is not completely clear how the monthly drought indices are correlated to ANNUAL crop yield. Months in Figures 7-9 begin in October so is that the October prior to the crop yield year? Also, can you explain what correlations of drought index to crop yield in say, December, actually means. By December all of the growing season and harvest is finished so what does this physically mean? Or, is this the December prior to the crop yield year? Please clarify some of these issues.

4. Line 147: The "atmospheric humidity demand" is only considered if using a Penman-Monteith or other physical evaporative demand formula. Palmer's original paper on used temperature and no humidity. I suggest changing just to "atmospheric demand".

5. Figures 2 and 3: Please clarify in the captions that the box plots are showing the distributions for all districts or provinces.

---

## Referee Comment (RC3) · Anonymous Referee #3 · 29 Mar 2019

The paper deals with the study of impact of drought on two representative rainfed crops in Spain (wheat and barley) through the analysis of the droughts events over this area from several drought indices, both multi-scalar and uni-scalar ones. This is an interesting task that has been assessed by other authors and the results of the present study must be put in those context. Some of these papers are:

Ribeiro, A.F.S., Russo, A., Gouveia, C.M. et al. (2018) "Modelling drought-related yield losses in Iberia using remote sensing and multiscalar indices" Theor. Appl. Climatol. 10.1007/s00704-018-2478-5.

Gouveia, Célia & Trigo, Ricardo & Beguería, Santiago & Vicente-Serrano, Sergio. (2016) "Drought impacts on vegetation activity in the Mediterranean region: An assessment using remote sensing data and multi-scale drought indicators" Global and

Planetary Change. 10.1016/j.gloplacha.2016.06.011, in a more general context.

Authors must be careful with some typo errors. For example: SYRS formulae or mistakes on citation as "by Lobell and Asner, (2003), Lobell et al., (2011) and Potopová et al., (2015)." (commas exceed). Also in the first paragraph of page 6 (line 186), it is not clear how many correlations were obtained, why 120?

Assessing these minor revisions, I recommend accept this paper, due its interest, if all these changes are made.

---

## Author Comment (AC1) · 7 May 2019

Authors sincerely thank to the three anonymous Referees for their reviews, constructive comments and positive feedback in general. We hope that they find the responses satisfactory.

Anonymous Referee #1

This article is interesting by combining data from different drought indices and yield of two crops at district level in region with water insufficiency. The methods used for statistical analysis of data sets are reasonable and provide an original contribution to the regional crop-loss and drought assessment in Spain. I have a few comments that need clarification before the paper can be considered for publication:

[Figure]

- Line 127: Please re-write SYR as SYRS. Thank you so much for noticing the typo. It has been changed.

Morover, the short name of the SYRS (standardised yield residuals series) was firstly introduced by Potopova et al. 2015.

Authors thank the notation made by Referee and changed the phrase as follow:

This methodology has been applied in other similar studies (Chen et al., 2016; Tian et al., 2018). First announced as 'SYRS' by Potopová et al., (2015), the full procedure of is described by Lobell and Asner, (2003) and Lobell et al., (2011).

- Please include in sec 2.1 as example plots with temporal trends of wheat and barley (t/ha) and plots with temporal evolution of SYRS of wheat and barley during the study period.

We have added a new supplementary figure as Reviewer suggests illustrating the temporal evolution of SYRS for both crops and the temporal trend of yields (t/ha) in a selected province and an agricultural district from the same province. Boxplots from Figure 2 already illustrate the temporal evolution of yields as a summary of all the provinces and districts analysed.

Thank you so much for the suggestion.

Please, see Supplementary Fig 1. in the new manuscript attached (page n°36).

Please also note the supplement to this comment:
https://www.nat-hazards-earth-syst-sci-discuss.net/nhess-2019-1/nhess-2019-1-AC1-supplement.pdf

---

## Author Comment (AC2) · 7 May 2019

Authors sincerely thank to the three anonymous Referees for their reviews, constructive comments and positive feedback in general. We hope that they find the responses satisfactory.

Anonymous Referee #2

This manuscript examines the spatial and temporal relationships of crop yield to a suite of drought indices in Spain. Two different crop yield datasets are used and seven drought indices are used including the multi-scalar SPEI, SPI, and SPDI and the uni-scalar PDSI, z-index, PHDI, and PMDI. Main findings were that then multi-scalar drought indices were most strongly related to crop yield and that spatial patterns varied

depending on time of year and drought index time scale. This paper is well written and of interest to the community. Overall, I did not find any major issues with the paper, but a few parts of the methods and analysis do need a bit more clarification. Once these methods are clarified the paper should be accepted for publication.

1. Figures 3-5 show the strongest correlations to crop yield but don't discuss the ending month of the strongest correlations. I realize month is discussed in other figures but I don't think it specifically highlights the ending month of those maximum correlations. Please clarify how the ending month is accounted for. An additional figure showing the spatial distribution of the ending months associated with max correlations could be useful.

In figures 3-5 we only attend to the magnitude of correlations stressing the difference of these among drought indices. The monthly response of yields to drought is summarized in the results obtained from the PCA (Figures 7 and 8). The PCA was performed using the maximum monthly correlations (pp. 212-214) and the comment of the temporal relationship between yields and climate is covered in point 3.3 and more specifically by crop and spatial scale in points 3.3.1 and 3.3.2. In discussion section, we also refer to the seasonality observed in the results and relate it with the phenology of the cultivations (pp. 488 – 495). Thanking the suggestion, authors think that adding supplementary figures showing the same maximum monthly correlations does not offer additional information beyond the one shown in the summary graph from the PCA.

2. Throughout the paper the "impact of drought" on crops is mentioned several times. However, all data for the period is used which includes droughts, wet years, and average years. I would think additional analysis would be needed to justify that wording that looks specifically at drought episodes. I would suggest somehow rephrasing that throughout the manuscript to more of a general "climate" impact of droughts as drought events are never discussed.

Authors appreciate this comment raised by the Referee and would like to offer their

opinion at respect. As Referee points out, it is true that we have not conducted an analysis of either a particular drought event or its impact on yields. Instead, we evaluated the capacity of different drought indices to detect changes on rainfed crop productions. Fluctuations on rainfed crop yields may respond to different causes such as plagues however, water availability is the main constraining factor. We aimed to assess how good or "less-good" each index analysed performs, and ultimately define the most appropriate index for drought monitoring purpose. In this paper, the term 'impact of drought' is referred succinctly in the context of the strong relationship found between the drought indices and the rainfed yields. We appreciate this comment as we think it is important to clarify it, for this reason, we have changed the use of the term 'impact of drought' for 'impact of climate' in those parts of the text where no reference to the strong relationship found between drought indices, and yields is made.

3. It is not completely clear how the monthly drought indices are correlated to ANNUAL crop yield. Months in Figures 7-9 begin in October so is that the October prior to the crop yield year?

Harvesting season ends in late July (in some regions this season extents until the firsts weeks of August), while sowing season starts (in general) in October. We made our climatic series start in October matching the beginning of the growing season. Thus, annual yields are related with climate starting in October from the previous year if it is consider January as the starting month.

Also, can you explain what correlations of drought index to crop yield in say, December, actually means. By December all of the growing season and harvest is finished so what does this physically mean? Or, is this the December prior to the crop yield year? Please clarify some of these issues.

In line with the previous response, December values correspond to the first stages of growth in both crops, when wheat and barley rainfed cultivations are more sensitive to precipitation shortages. This is observed in figures 7/8 and 9 as maximum monthly correlations are recorded in May at 4/5-month time-scales (corresponding to Dec-January climate conditions).

4. Line 147: The "atmospheric humidity demand" is only considered if using a Penman-Monteith or other physical evaporative demand formula. Palmer's original paper on used temperature and no humidity. I suggest changing just to "atmospheric demand".

We take note of the suggestion in the manuscript. Thank you so much.

5. Figures 2 and 3: Please clarify in the captions that the box plots are showing the distributions for all districts or provinces.

We have added it to the caption. Thank you.

Fig. 3. Box plots showing the distribution of the strongest correlation coefficients found between drought indices and wheat and barley yields at the agricultural district (a and b) and provincial (c and d) scales, for all districts and provinces analysed. The solid black line shows the median, the white asterisk shows the mean, and the dashed red lines show the $p < 0.05$ significance level.

Please also note the supplement to this comment:
https://www.nat-hazards-earth-syst-sci-discuss.net/nhess-2019-1/nhess-2019-1-AC2-supplement.pdf

**Supplement:**

[revised manuscript text omitted]

**Figures**

---

## Author Comment (AC3) · 7 May 2019

- Authors sincerely thank to the three anonymous Referees for their reviews, constructive comments and positive feedback in general. We hope that they find the responses satisfactory.

Anonymous Referee #3

The paper deals with the study of impact of drought on two representative rainfed crops in Spain (wheat and barley) through the analysis of the droughts events over this area from several drought indices, both multi-scalar and uni-scalar ones. This is an interesting task that has been assessed by other authors and the results of the present study must be put in those context. Some of these papers are:

[Figure]

Ribeiro, A.F.S., Russo, A., Gouveia, C.M. et al. (2018) "Modelling drought-related yield losses in Iberia using remote sensing and multiscalar indices" Theor. Appl. Climatol. 10.1007/s00704-018-2478-5.

Gouveia, Célia & Trigo, Ricardo & Beguería, Santiago & Vicente-Serrano, Sergio. (2016) "Drought impacts on vegetation activity in the Mediterranean region: An assessment using remote sensing data and multi-scale drought indicators" Global and Planetary Change. 10.1016/j.gloplacha.2016.06.011, in a more general context.

- Authors thank very much for the contribution of this Referee on suggesting these references that enrich the corpus of the manuscript. We took note and added new references to contextualize the study and the results in both sections, Introduction (pp. 55-58) and Discussion (pp. 459-460; 489-491; 503-505). Thank you.

Authors must be careful with some typo errors. For example: SYRS formulae or mistakes on citation as "by Lobell and Asner, (2003), Lobell et al., (2011) and Potopová et al., (2015)." (commas exceed).

- We have carefully revised the citations, removed the commas and corrected the typo error mentioned. Thank you so much.

Also in the first paragraph of page 6 (line 186), it is not clear how many correlations were obtained, why 120?

- Authors really appreciate this note because there was a mistake in the original manuscript. The correct number of correlations obtained is 140 (1 correlation obtained for each of the 10 months analysed * the 14 time-scales considered).

Assessing these minor revisions, I recommend accept this paper, due its interest, if all these changes are made.

- Thank you so much.

Please also note the supplement to this comment:

[Figure]

https://www.nat-hazards-earth-syst-sci-discuss.net/nhess-2019-1/nhess-2019-1-AC3-supplement.pdf

**Supplement:**

[revised manuscript text omitted]

(*) correlations are significant at p < 0.05

Supplementary Table 2. Relationship between provincial and agricultural district data, aggregated at provincial scale, for barley cultivation for the common period 1993−2014.

| Codes | Provinces | r |
|-------|-----------|------|
| 1 | Álava | 0.11 |
| 2 | Albacete | 0.2 |
| 10 | Cáceres | 0.48* |
| 11 | Cádiz | 0.32* |
| 12 | Castellón | -0.14 |
| 13 | Ciudad Real | 0.28 |
| 14 | Córdoba | 0.54* |
| 15 | A Coruña | -0.09 |
| 16 | Cuenca | 0.88* |
| 17 | Girona | 0.08 |
| 18 | Granada | 0.51* |
| 19 | Guadalajara | 0.86* |
| 22 | Huelva | 0.57* |
| 26 | La Rioja | 0.76* |
| 31 | Navarra | 0.01 |
| 41 | Sevilla | -0.35* |
| 43 | Tarragona | 0.88* |

(*) correlations are significant at p < 0.05

[Figure]

Supplementary Fig. 1. Example of temporal trends of provincial and agricultural district yields of wheat (a, d) and barley (b, e) in the province of Cáceres and the district Navalmoral de la Mata (Cáceres) and the temporal evolution of the SYRS at both scales (c, f) for the available period of time in each case. Red line represents the fitting of a quadratic function. Dashed black line represents the threshold 0-value.

[Figure]

Supplementary Fig. 2. Spatial distribution of regions where significant differences (dark grey) and non significant differences (light grey) were found in the t-tests.

[Figure]

Supplementary Fig. 3. Monthly mean AED conditions in the agricultural districts where wheat was cultivated, classified into principal components (C1 and C2) for the period 1993−2015. The red dot shows the mean, and the black line shows the median.

[Figure]

Supplementary Fig. 4. As for Supplementary Fig. 3, but for the monthly mean precipitation.

[Figure]

Supplementary Fig. 5. As for Supplementary Fig. 3, but for the monthly mean maximum temperature.

[Figure]

Supplementary Fig. 6. As for Supplementary Fig. 3, but for the monthly mean minimum temperature.

[Figure]

Supplementary Fig. 7. As for Supplementary Fig. 3, but for the monthly mean hydroclimate balance.

[Figure]

Supplementary Fig. 8. Monthly mean AED conditions in the agricultural districts where barley was cultivated, classified into principal components (C1 and C2) for the period 1993–2015. The red dot show the mean, and black line shows the median.

[Figure]

Supplementary Fig. 9. As for Supplementary Fig. 8, but for the monthly mean precipitation.

[Figure]

Supplementary Fig. 10. As for Supplementary Fig. 8, but for the monthly mean maximum temperature.

[Figure]

Supplementary Fig. 11. As for Supplementary Fig. 8, but for the monthly mean minimum temperature.

[Figure]

Supplementary Fig. 12. As for Supplementary Fig. 8, but for the monthly mean hydroclimate balance.